# Resiniferatoxin: Nature’s Precision Medicine to Silence TRPV1-Positive Afferents

**DOI:** 10.3390/ijms242015042

**Published:** 2023-10-10

**Authors:** Arpad Szallasi

**Affiliations:** Department of Pathology and Experimental Cancer Research, Semmelweis University, 1083 Budapest, Hungary; szallasi.arpad@semmelweis.hu

**Keywords:** resiniferatoxin, capsaicin, TRPV1 receptor, cancer pain, osteoarthritic pain, incontinence

## Abstract

Resiniferatoxin (RTX) is an ultrapotent capsaicin analog with a unique spectrum of pharmacological actions. The therapeutic window of RTX is broad, allowing for the full desensitization of pain perception and neurogenic inflammation without causing unacceptable side effects. Intravesical RTX was shown to restore continence in a subset of patients with idiopathic and neurogenic detrusor overactivity. RTX can also ablate sensory neurons as a “molecular scalpel” to achieve permanent analgesia. This targeted (intrathecal or epidural) RTX therapy holds great promise in cancer pain management. Intra-articular RTX is undergoing clinical trials to treat moderate-to-severe knee pain in patients with osteoarthritis. Similar targeted approaches may be useful in the management of post-operative pain or pain associated with severe burn injuries. The current state of this field is reviewed, from preclinical studies through veterinary medicine to clinical trials.

## 1. Resiniferatoxin: A 2000-Year History in a Snapshot

For pharmacologists, natural products provide a window of opportunity for discovering new therapeutic targets. A prominent example is resiniferatoxin (RTX), the active principle in *Euphorbium*, a drastic medicinal plant resin known since Roman times [1]. *Euphorbium* is the dried latex of *Euphorbia resinifera* Berg, a large, leafless cactus-like perennial [2]. The plant is a native of the Anti-Atlas Mountain of Morocco. Although the complete chemical synthesis of resiniferatoxin is now resolved [3], the compound is still isolated in the traditional way from plants cultivated in Arizona deserts. The latex (milky juice) of *E*. *resinifera* Berg has been collected since ancient times by wounding the stems of the plants. 

The etymology of *Euphorbia* remains elusive. In Greek, it means “well-fed”, which may refer to the fattened look of the plant. Others translate it as “good fodder”, since famished camels are known to eat cactus-like succulents due to a lack of tastier alternatives. 

Despite these uncertainties, a consensus appears to exist among historians that *Euphorbium* was really named after the Roman physician Euphorbius (not to be confused with Euphorbus, one of Troy’s finest warriors), who used *Euphorbium* to treat the arthritic pains of the Emperor Augustus [4]. In hindsight, this medical use of *Euphorbium* can be considered as scientifically solid, since RTX has just been granted breakthrough therapy designation by the US Food and Drug Administration for osteoarthritic pain [5].

One hundred sixty-five years ago, Robert Virchow, in his “*Reiztheorie*” (inflammation theory), predicted a causative connection between inflammation and cancer [6]. Subsequently, a close relationship was postulated between the inflammatory and tumor-promoting activities of natural products [7]. The prototypical tumor promoter 12-O-tetradecanoylphorbol-13-acetate (TPA) is indeed a very potent inflammatory agent in mouse skin. Therefore, the mouse ear erythema assay was routinely used to detect tumor-promoting compounds in natural product extracts. 

In 1975, an extremely irritant compound was discovered in the soap of *Euphorbia resinifera* Berg [8], named after its source: resiniferatoxin (RTX). However, unlike TPA, RTX did not promote the formation of tumors [9]. Furthermore, RTX was only marginally active at activating the main phorbol ester target, protein kinase-C [10].

In fact, tumor-promoting phorbol esters and RTX are structurally similar, with a critical difference at the C20 position: whereas tumor-promoting phorbol esters carry a free OH group at this position, RTX is, by contrast, esterified with homovanillic acid (Figure 1).

The turning point in the two-thousand-year-old history of *Euphorbium* came in 1989, when RTX was identified as an ultrapotent analog of capsaicin (Table 1), the pungent ingredient in hot chili peppers [11]. Subsequently, specific binding of [^3^H]RTX provided the first biochemical proof for the existence of the long sought-after capsaicin receptor [12]. Based on the shared chemical motif that is essential for the bioactivity of both capsaicin and RTX (Figure 1), this receptor was termed the vanilloid receptor-1, or briefly, the VR1 receptor [13]. The term “vanilloid” survives in the somewhat cumbersome name of the cloned capsaicin receptor: Transient Receptor Potential, Vanilloid Subfamily Member-1 (TRPV1) [14,15]. 

The phenotype of the TRPV1 null mouse (attenuated thermal hyperalgesia under inflammatory conditions [20,21]) triggered tremendous interest in developing small-molecule TRPV1 antagonists as a novel class of analgesic drugs [22]. It took less than a decade for the first of these antagonists to enter Phase I clinical trials after the cloning of TRPV1 [22]. However, as of today, none of these compounds have reached Phase III trials. Some failed due to a lack of clinical efficacy, whereas others had to be withdrawn due to side effects, most notably hyperthermia and burn injuries [22,23]. These problems with TRPV1 antagonists have rekindled interest in TRPV1 agonists, such as capsaicin and RTX, as an alternative means to block (desensitize) TRPV1-expressing sensory neurons. 

The clinical value of high-dose capsaicin creams [24] and site-directed injections [25] has been reviewed extensively. Here, we focus on RTX as an “improved” capsaicin analog. Of the 1062 RTX-related publications in PubMed, approximately 200 were selected for critical review, ranging from molecular mechanisms of action through preclinical models to clinical trials. Every clinical study with RTX in pain patients (eight publications) and bladder disorders (19 papers) is included. Of the 638 preclinical studies, only those were selected that paved the way for human studies. 

## 2. The Mechanism of Action of RTX and Capsaicin: Similarities and Differences

The molecular target for capsaicin, now known as TRPV1, was cloned in 1997 [15]. In 2021, this discovery earned a Nobel Prize in Physiology and Medicine (shared with Ardem Patapoutian) for David Julius. Whoever has sampled hot chili pepper knows from personal experience that capsaicin evokes a “hot”, burning sensation in the human tongue. Thus, it was hardly unexpected that TRPV1 turned out to be a shared target for heat and capsaicin [15]. In fact, TRPV1 is the founding member of the functional group of temperature-sensitive TRP channels, the so-called “thermoTRPs” [26,27]. Combined, these channels cover a broad range of temperatures, from noxious hot to harmful cold. Both experimental animals [20,21] and people [28] with non-functioning TRPV1 channels show deficits in noxious heat sensation. Furthermore, in clinical studies, the pharmacological blockade of TRPV1 by small-molecule antagonists caused burn injuries as adverse effects [23]. Thus, the central role of TRPV1 in noxious heat sensation is firmly established. 

Of note, most recently, a human missense *TRPV1* variant (K710N) has been discovered that, when expressed in knock-in mice using CRISPR/Cas9, rendered the animal insensitive to nerve injury-induced hyperalgesia, while leaving noxious heat sensation intact [29]. This finding raises the possibility that such analgesic TRPV1-targeting drugs may be developed that do not cause burn injuries as a dose-limiting side effect.

Transient Receptor Potential (TRP) channels comprise a superfamily of 28 members (27 in humans, where TRPC2 is only a pseudogene) [30,31,32,33]. The name TRP originates from the phenotype of a mutant fruit fly. The eye of wild-type *Drosophila* responds to constant light with a sustained current. By contrast, the eye of this mutant ceased to function after a transient response even when exposed to light [34], hence the term “transient receptor potential.” 

Based on sequence homology, the TRP superfamily has been divided into six subfamilies: ankyrin, canonical, melastatin, polycystin, mucolipin, and vanilloid [30,31,32,33]. The capsaicin receptor TRPV1 is the founding member of the vanilloid TRP subfamily (TRPV1 to TRPV6). 

The biology and pharmacology of TRPV1 have been reviewed extensively [35,36,37,38]. Here, it suffices to mention that in sensory afferents, TRPV1 is a “promiscuous” integrator of noxious stimuli [35], ranging from temperature [15] and changes in pH [39], through toxins in spiders [40,41] and jellyfish [42], to mechanical stimuli and voltage [43]. Furthermore, TRPV1 is a downstream target for pain-producing substances such as bradykinin [44]. The activity of TRPV1 is also regulated by posttranslational modification of the channel protein. For example, phosphorylation by protein kinase-C [45,46,47] and neuron-specific cyclin-dependent kinase-5 (Cdk5) [48,49] sensitizes TRPV1 to agonists, whereas cAMP-dependent protein kinase regulates TRPV1 desensitization [50]. 

In plasma membrane, TRPV1 functions as a cation channel with limited selectivity for Ca^2+^. TRPV1 was also detected in subcellular locations, such as the mitochondria, with less understood functions [51]. TRPV1 predominantly exists as homotetramer [52], but it can also form heteromultimers with other TRP channels like TRPA1 [53]. TRPV1 splice variants with dominant negative functions were also identified [54,55]. Recently, cryo-electron microscopy has provided important insights into channel activation by TRPV1 agonists [56,57,58,59]. RTX binding to each of the four subunits of functional mouse TRPV1 contributes virtually the same activation energy to destabilize the closed conformation [59].

Functional TRPV1 is expressed, albeit at much lower levels than in sensory neurons, in brain nuclei [60,61], as well as in non-neuronal cells, including keratinocytes [62], endothelial cells [63], and vascular smooth muscle [64]. The physiological role of these TRPV1-expressing cells is subject to intensive research. 

Capsaicin is unique among sensory irritants in that the initial excitation that it evokes is followed by a lasting refractory state, traditionally termed “desensitization”, in which the previously excited neurons are unresponsive to a broad range of unrelated painful stimuli [65,66,67,68,69]. Capsaicin-sensitive neurons have cell bodies in sensory ganglia (dorsal root and trigeminal) and give rise to thin, unmyelinated bi-polar axons [69]. The peripheral terminals of these neurons are sites of release for a variety of pro-inflammatory neuropeptides (e.g., calcitonin gene-related peptide, CGRP) that, in turn, trigger the biochemical cascade known as neurogenic inflammation (efferent function) [65,67]. The central fibers terminate in the dorsal horn of the spinal cord, where they make synapses at second-order neurons and transmit noxious information into the central nervous system (afferent function) [67,69]. In addition to their primary role in detecting potentially harmful environmental stimuli (such as noxious heat), capsaicin-sensitive neurons are also involved in body temperature regulation [70,71,72]. 

Unlike capsaicin that evokes a rapid, burst-like Ca^2+^ influx through TRPV1 [73], the RTX-induced current is slow and sustained [74]. This may explain to some degree why RTX is much less irritating than capsaicin at doses needed to achieve clinically useful desensitization (Table 1) [11,69]. The sustained RTX-induced current still delivers Ca^2+^ in quantities sufficient to render the nerves non-functional [75]. RTX is a bulkier molecule than capsaicin (compare structures in Figure 1). The vanilloid binding site on TRPV1 is intracellular [76]. Therefore, lipophilicity is an important factor in determining the kinetics of agonist-induced TRPV1 activation. In fact, the pungency of vanilloids correlates with their lipophilicity [77], and RTX can induce loss of the plasma membrane in TRPV1-expressing neurons [78].

Per definition, desensitization by RTX is reversible. However, RTX can also kill sensory neurons [79,80,81], an irreversible effect. The toxic action of RTX is thought to be due to a combination of Ca^2+^ overload and osmotic injury [75,79]. As we will see below, both reversible desensitization and permanent ablation of sensory neurons can be exploited for therapeutic purposes, most notably for pain control. Of note, desensitization to RTX may last for several months and is repeatable by a second RTX administration [82]. 

The molecular mechanisms by which RTX can cause such lasting desensitization are only beginning to be understood. Unfortunately, many authors use the term “desensitization” rather loosely, not making a clear distinction between desensitization and inactivation or even axotomy that occur through Ca^2+^ overload [83]. 

There is good evidence that the phenotype of sensory neurons can change from pro-algesic to analgesic after RTX treatment. This change was referred to as “vanilloid-induced messenger plasticity” [84]. For example, substance P, a neuropeptide produced by sensory neurons that plays an important role in pain transduction, is down-regulated after RTX challenge [85], whereas galanin (a neuropeptide with analgesic activity) is upregulated [86]. More recently, widespread RTX-induced change in neuronal gene expression has been described [87]. Importantly, in rat [88], dog [89], or human [90] tissue biopsies, no significant morphological changes were noted at the light or electron microscopy levels following local RTX administration. (It should be noted here that these tissue biopsies sampled only the peripheral terminals of TRPV1-expressing neurons and not the cell bodies. In the cell bodies, swollen mitochondria can be detected by electron microscopy following capsaicin administration as a hallmark of capsaicin damage [91]. It is possible that RTX can also induce a similar swelling of the mitochondria.)

In rodents, capsaicin exerts a tri-phasic effect on blood pressure regulation: (1) an initial period of hypotension, followed by (2) a more sustained phase of blood pressure elevation that progresses into (3) another drop in blood pressure [92]. The initial hypotensive effect is attributed to the triggering of the Bezold–Jarisch reflex [92], also known as the pulmonary chemoreflex. In the second phase, capsaicin directly constricts blood vessels by activating TRPV1 expressed in vascular smooth muscle [64]. Less clear is the mechanism of the third phase: this may be a late effect of neuropeptide (for example, CGRP) release or a direct capsaicin action on endothelial cells expressing TRPV1 [63].

The Bezold–Jarish reflex is a triad that also includes effects on heart rate (bradycardia) and respiration (bradypnea). Activation of the Bezold–Jarisch reflex limits the dose of capsaicin that can be given systematically. Therefore, only partial desensitization of the neurogenic inflammatory response can be achieved by a single s.c. dose of capsaicin [11,64]. In order to attain full desensitization, capsaicin needs to be given in increasing doses over a period of several days [93]. By contrast, RTX does not evoke the Bezold–Jarisch reflex, although it renders the reflex pathway unresponsive to a subsequent capsaicin challenge [94]. Consequently, a full desensitization of neurogenic inflammation can be achieved by means of a single s.c. RTX injection (Figure 2) [11]. In other words, RTX has a better, wider therapeutic window than capsaicin (Figure 2). However, at supratherapeutic doses, s.c. RTX (similar to capsaicin [95]) may cause skin ulcers by eliciting scratching behavior in experimental animals.

## 3. RTX and Bladder Disorders: Experimental Models

TRPV1 is abundantly expressed in bladder C-fibers [96,97,98]. In the bladder mucosa, TRPV1-expressing nerve endings terminate in the suburothelial space, where they are apposed to basal urothelial cells. Functional TRPV1 was also demonstrated in the urothelium [99,100]. It was speculated that urothelial TRPV1 may sense changes in urine pH and/or osmolarity [101]. In human urothelium, TRPV1 responds to capsaicin and heat with ATP release [102]. Rat urothelial cells also respond with ATP release to stretch and increased hydrostatic pressure [103]. The stretch-evoked ATP release was greatly attenuated in the bladder of *Trpv1* (-/-) mice [104]. Increased TRPV1 expression was demonstrated in the urothelium of patients with overactive bladder [105,106] and painful bladder/interstitial cystitis [107]. These observations indicate that TRPV1 participates in both normal bladder function and pathology. 

Rats desensitized to capsaicin have a greatly distended urinary bladder [108,109]. *Trpv1* (-/-) mice also exhibit increased bladder capacity, associated with frequent small-volume (“spotty”) voiding contractions [104,110]. These findings were interpreted to imply an important role for C-fibers in the micturition reflex. However, TRPV1 knockout mice did not differ in non-voiding bladder contractions from their wild-type littermates, nor did they show non-voiding contractions during bladder filling [104]. Furthermore, the small-molecule TRPV1 antagonist GRC-6211 had no effect on bladder functions in control animals, though it blocked bladder overactivity in response to cystitis or spinal cord injury (SCI), a model of neurogenic bladder [111]. Thus, one may argue that C-fibers do not drive the micturition reflex under physiological conditions, but they assume control in the inflamed or neurogenic bladder. In accord, TRPV1-null animals lack bladder overactivity during experimental cystitis [112].

Of note, in rodents, the density of TRPV1-positive bladder afferents decline with age [113]. If this observation holds true for humans, intravesical vanilloid therapy may not be indicated in the elderly. 

In awake, freely moving rats, intravesical RTX (0.1 to 10 nmol) induced lasting desensitization with near complete recovery within 7–14 days [114]. The initial discomfort (e.g., lower abdominal licking [115]) could be eliminated by tetracaine [116]. At higher RTX concentrations (up to 100 nmol), the desensitization lasted longer (4 weeks or more) but still returned after 8 weeks [117]. Using continuous cystometry, RTX was at least a thousand-fold more potent than capsaicin in facilitating micturition [118].

In rats, intravesical RTX prevented the development of neurogenic cystitis induced by the pseudorabies (PRV)-Bartha virus [119]. 

CSI rats (chronic spinal injury at the level of the eighth and ninth thoracic vertebra) represent a model of neurogenic bladder. CSI rats develop rhythmic bladder contractions. Intravesical RTX (1–10 µM) blocks these contractions, implying a therapeutic potential in patients with neurogenic bladder [120]. 

At desensitizing concentrations (up to 1 µM), intravesical RTX did not increase *fos* expression (a neurochemical marker of nociception) in rat sensory neurons [117,121]. RTX, however, did deplete substance P (SP) and CGRP from bladder afferents [88,122]. Importantly, SP and CGRP levels returned to control by 4 weeks after RTX administration [88]. Ultrastructural (EM) studies ruled out axon degeneration [88]. Taken together, the rat experiments imply that intravesical RTX induces a long-lasting but reversible desensitization of the TRPV1-expressing bladder afferents, with no or minimal discomfort. 

There is no rodent model of interstitial cystitis. Intravesical RTX (0.1 µM for 30 min) was tested in two control cats and another two with feline interstitial cystitis [123,124]. RTX increased bladder capacity in the interstitital cystitis, but not the healthy, animals. 

In summary, intravesical RTX is an effective, well-tolerated, and safe intervention in experimental models of neurogenic bladder and cystitis-associated discomfort. Preliminary evidence also suggests a therapeutic value for RTX in interstitial cystitis. 

## 4. RTX and Bladder Disorders: Clinical Studies

In 15 subjects with normal detrusor function, intravesical RTX (10^−8^ M for 30 min) did not evoke an unpleasant burning sensation, nor did it change the bladder capacity at which the first desire to void occurs [125]. The same RTX concentration, however, did increase the bladder capacity by an average of 175 mL in patients with bladder hyperactivity [125]. 

In a second study, RTX (100 mL of a 50 nM or 100 nM solution in 10% EtOH) was administered via a catheter into the bladder of seven patients with overactive bladder [126]. Three patients had multiple sclerosis, and one patient each had spinal cord injury, extradural abscess, transverse myelitis, and stroke, respectively. No patient indicated intense pain, and no change in vital parameters was noted [126]. Importantly, the temporary deterioration in bladder functions often observed with intravesical capsaicin was absent in the RTX-treated patients. In five patients, the average urinary frequency decreased from 10 to 26 times per day to 6 to 12 times per day (a 30–50% improvement) [126]. Furthermore, three incontinent patients became dry, with the resumption of normal urinary frequency for 3 months after RTX treatment. The mean cystoscopic capacity (MCC) increased by 76 to 596 mL, indicating significant differences in patient responses. Furthermore, one patient did not respond to the treatment at all. Biopsies taken from the bladder of patients following RTX administration revealed minimal chronic inflammation but no ultrastructural (EM) damage [90]. 

In 14 patients with detrusor hyperreflexia, intravesical RTX (50 or 100 nM for 30 min) was well tolerated [82]. In 9 of the 14 study subjects, RTX improved or abolished incontinence. The mean urinary frequency decreased from 14.2 to 10.3 times per day, whereas the MCC increased in the range of 182 to 330 mL. The ice-water test became negative in eight patients [82]. The beneficial RTX effect was long-lasting, still present in seven patients after 12 months of treatment. Intravesical RTX (10 µM) even increased cystomanometric capacity by an average of 217 mL in seven patients with detrusor hyperreflexia refractory to intravesical capsaicin [127]. 

Five prospective, randomized clinical trials involving 18, 24, 35, 36, and 40 patients with neurogenic bladder due to spinal cord injury confirmed the therapeutic value of intravesical RTX in this patient population [128,129,130,131,132]. In a meta-analysis of four clinical trials involving a total of 288 patients with multiple sclerosis, intravesical RTX was superior to placebo [133]. 

Intravesical RTX was also effective in patients with idiopathic detrusor instability refractory to anticholinergics [134,135]. However, in a second study with 58 patients, RTX was no better than placebo [136]. 

In 12 patients with benign prostatic hyperplasia (BPH)-associated lower urinary tract syndrome (LUTS), intravesical RTX (50 nM) decreased the mean urinary frequency from 15.2 to 10.8 times per day. Importantly, these patients became free of urge incontinence [137]. 

In a pilot study with five female interstitial cystitis (IC) patients, intravesical RTX (10 nM) was reported to reduce both nocturia and bladder discomfort [138]. In three Japanese patients with IC, intravesical RTX achieved some improvement in maximal voided volume [139]. In a cohort of 13 patients with refractory IC, 10 nM RTX administered once a week for 4 weeks resulted in excellent clinical response in two patients, some improvement in five patients, and no response in five patients [140]. By contrast, in a randomized clinical trial involving 163 interstitial cystitis patients, intravesical RTX was well tolerated but lacked clinical efficacy [141]. A meta-analysis of nine eligible clinical trials with female painful bladder syndrome/IC patients also failed to detect any benefit for intravesical RTX [142]. 

In 48 patients with catheter-related bladder discomfort, intravesical RTX relieved pain [143]. 

In summary, intravesical RTX is an effective treatment with good tolerability in a subset of patients with idiopathic destrusor instability, neurogenic bladder, and BPH-associated LUTS. In interstitial cystitis, RTX may also help selected patients, but it is unclear how these patients can be identified. The lack of standardized intravesical RTX preparations hinders the interpretations of the negative results. 

Of note, in the bladders of patients with idiopathic [144] or neurogenic detrusor overactivity [106], increased TRPV1 expression was demonstrated. Therefore, one may argue that TRPV1 immunostaining of cystoscopic bladder biopsies may help select patients who may benefit from intravesical RTX therapy. 

The molecular mechanism by which TRPV1 expression is increased in the bladder under pathological conditions is poorly understood. One possible explanation involves nerve growth factor (NGF). The *TRPV1* gene promoter contains an NGF-responsive element [145], and the urine of patients with detrusor instability shows elevated NGF levels [146]. 

## 5. RTX for Pain Relief: Animal Studies

In the Bennett model of neuropathic pain, loose ligatures are placed on the sciatic nerve of the rat. These animals develop both mechanical and thermal hyperalgesia. The thermal hyperalgesia can be made quantitative by measuring the withdrawal latency of the affected hind paw from the hot plate. In this model, systemic (s.c.) RTX treatment was studied in two protocols (Figure 3) [69]. In the first protocol, RTX had been given preventively before sciatic nerve surgery was performed. In the second protocol, the operated animals were injected with RTX when the full thermal hyperalgesia had developed (Figure 3). Systemic RTX administration (100 µg/kg s.c.) prevented the development of thermal hyperalgesia in the first protocol and reversed the thermal hyperalgesia in the second protocol (Figure 3) [69]. Thus, systemic RTX desensitization is a powerful tool to achieve analgesia in experimental animals. In patients, however, this approach cannot be pursued due to foreseeable adverse effects. As discussed later, in humans, targeted (site-specific) RTX injections provide an attractive alternative to systemic desensitization. RTX can be injected around the sensory nerve, into the sensory ganglion, or directly into the painful location. A prominent example is intra-articular RTX injection into to knee joint of patients with moderate-to-severe osteoarthritis [5]. Lastly, RTX can be administered intrathecally.

In adult Sprague–Dawley rats, inflammatory pain can be induced by injecting carrageenan (6 mg in 150 µL) into the hind paw. In these animals, perineural RTX (25 to 250 ng in a volume of 50 µL around the sciatic nerve) significantly increased the hind-paw heat withdrawal latency at the ipsilateral carrageenan injection side [147,148,149,150]. At the highest RTX dose tested (250 ng), the analgesic effect was long-lasting, up to 6 months after treatment. Thereafter, the animals showed full recovery of the heat pain response. Importantly, the perineural RTX injection did not change the mechanical sensitivity of the animals (von Frey anesthesiometer), nor did it influence the motor coordination (rotarod response), indicating high selectivity for polymodal C-fibers [151]. Moreover, an electron microscopic study of the sciatic nerve found no evidence of C-fiber damage following perineural RTX (up to 1 µg) injection [152]. 

In a rat model of postoperative pain, perineural RTX prevented the development of incisional hyperalgesia [153]. Moreover, local RTX injection into the injured area produced lasting analgesia in a rat model of full-thickness burn injury [154]. 

The monoiodoacetate (MIA) model has become a standard for studying joint disruption in osteoarthritis. Various studies demonstrate that MIA evokes pain and avoidance behaviors in experimental animals that are similar to those reported in osteoarthritis patients. In rats, MIA (8 mg/50 µL injected into the knee joint) induces inflammation and damages cartilage. In rats with MIA-induced osteoarthritis, intra-articular RTX (0.0003% to 0.03%) increased paw withdrawal latency to radiant noxious heat and mechanical stimuli [155]. It also reduced the time that animals used for weight bearing on the contralateral limb. Moreover, intra-articular RTX ameliorated pain-related behavior in arthritic dogs (Figure 4) [156].

In rats, stereotactic unilateral RTX microinjection (20 or 200 ng in 4 µL) into the trigeminal ganglia blocked both the afferent (capsaicin-evoked eye-wiping response) and efferent (neurogenic inflammation detected by Evan’s blue extravasation) function of TRPV1-positive C-fibers [157]. The antinociceptive action of RTX was maintained for at least one year. Intra-ganglionic RTX (200 ng) permanently eliminated approximately 80% of the TRPV1-positive trigeminal ganglion neurons [157]. 

In Rhesus monkeys, RTX was microinjected into the trigeminal ganglia [158], whereas in pigs, it was administered around the lumbar L2/3 DRG using a CT-guided transforaminal approach [159]. In the monkeys, like in rats [160], intra-ganglionic RTX eliminated both the capsaicin-induced eye-wiping response and the neurogenic inflammatory response [158].

In pigs, periganglionic RTX (2000 ng) reduced TRPV1 mRNA by 66.3% compared to solvent control [159], with no neurological deficits or post-injection complications [161]. No gait abnormalities were noted in either species.

Severe cancer pain caused by bone metastasis usually affects more than one dermatome. Such generalized pain can be targeted by intrathecal RTX administration that reaches multiple ganglia via the cerebrospinal fluid [162]. In rats, intrathecal RTX (10 to 200 ng given by lumbar puncture) increased the withdrawal latency of hind paws to radiant noxious heat in a dose-dependent manner, with no effect on the front paws [162]. In a dose–response study, the ED_50_ for epidural RTX was determined as 265 ng [163]. At this dose, the increased latency to thermal stimulation continued for at least 20 days, the end of the study [163]. At a “supratherapeutic” RTX dose (5000 ng), two out of the six treated animals died [163]. 

In the hind paws, intrathecal RTX blocked carrageenan-induced thermal hyperalgesia. In the dorsal horn of the spinal cord, intrathecal RTX deleted most TRPV1/CGRP-positive nerve endings [164]. Furthermore, intrathecal RTX (1.9 µg/kg) prevented the development of tumor necrosis factor (TNF)-induced pain behavior in the rat [165].

Importantly, intrathecal RTX had no effect in normal mechanosensation (von Fey hair or pin prick) or motor coordination [81]. A second, independent study confirmed the permanent loss of TRPV1-positive neurons following intrathecal RTX administration [166]. 

In a murine model of cancer pain, intrathecal RTX produced lasting analgesia [167], similar to that observed after disruption of the *Trpv1* gene [168]. Of note, pancreatic cancer cells express TRPV1 both at the mRNA and protein levels [169]. If the TRPV1 on cancer cells is functional, RTX may suppress cancer pain and decrease tumor volume at the same time.

Lastly, intrathecal RTX ameliorated the signs of prostatodynia evoked by the injection of complete Freund’s adjuvant into the rat prostate [170]. 

These observations imply a therapeutic potential for (1) intraganglionic RTX for orofacial neuropathic hyperalgesia, (2) local/site-specific RTX injections for arthritic pain and burn injury-related pain, and (3) intrathecal RTX administration for cancer pain. 

## 6. Intrathecal RTX for Permanent Pain Relief in Companion Dogs with Bone Cancer

Large dogs are prone to developing osteosarcoma in their legs. These animals limp and exhibit behavioral signs of severe pain. In 20 dogs with intractable bone cancer pain, intrathecal RTX (1.2 µg/kg) was given under general anesthesia [171]. The most sensitive structure for intrathecal RTX is the TRPV1-expressing axon exposed to the cerebrospinal fluid [172]. After a period of 45–60 min, the animals were awakened and their vitals tested. The RTX-treated dogs showed a significant increase in mean arterial pressure (in the range of 79 to 131 mmHg) and heart rate (123 to 160 beats per min) [173]. These cardiovascular effects peaked at 1–2 h after RTX administration and then slowly returned to normal by 4 h. The dogs also demonstrated behavioral signs of discomfort (barking and panting), which disappeared after diazepam and hydromorphone treatment. For several hours, the treated animals showed hypothermia. 

Intravenous capsaicin is known to elevate blood pressure by increasing peripheral vascular resistance [92]. Smooth muscle cells in the wall of resistance arteries express TRPV1; the activation by capsaicin of this TRPV1 leads to vasoconstriction [64]. Although RTX may mimic this capsaicin action, blood RTX after intrathecal administration is unlikely to reach concentrations high enough to elevate blood pressure. 

The RTX-induced hypothermic response is easier to explain. Hypothermia is a well-known effect of systemic capsaicin or RTX administration [66,69]. Capsaicin microinjected into the brain (preoptic area) causes a drop in body temperature [174]. Intrathecal RTX probably reaches a capsaicin-responsive area in the CNS, responsible for the thermoregulation. 

The day after intrathecal RTX treatment, dogs appeared to be normal on physical examination [171]. The animals became ambulatory, and caregivers reported improved comfort levels (Figure 5). RTX, however, had no effect on the progression of the cancer. When the animal died, a full autopsy was carried out that did not find any morphological changes that may have been attributed to the RTX treatment [174].

In a prospective, randomized (RTX or standard-of-care), and blinded trial involving 72 companion dogs with bone cancer pain, RTX (1.2 µg/kg) was injected into the cisterna magna for forelimb tumors or the L5/6 lumbar interspace (lumbar puncture) for hind-limb cancers [174]. “Unblinding” was performed when the caregiver indicated too much pain. Dogs in the standard-of-care group were “unblinded” sooner and more often than in the RTX group (78% and 50%, respectively), indicative of clinically significant pain relief by RTX (Figure 5) [174].

Of note, three of fourteen beagle dogs died after intracisternal RTX (3.6 µg/kg) treatment [89]. This finding cautions about the dosing of centrally administered RTX.

In summary, intrathecal RTX is a well-tolerated and effective treatment of severe cancer pain in companion dogs, incentivizing the transition to human clinical trials [175,176,177].

**Figure 5 ijms-24-15042-f005:**
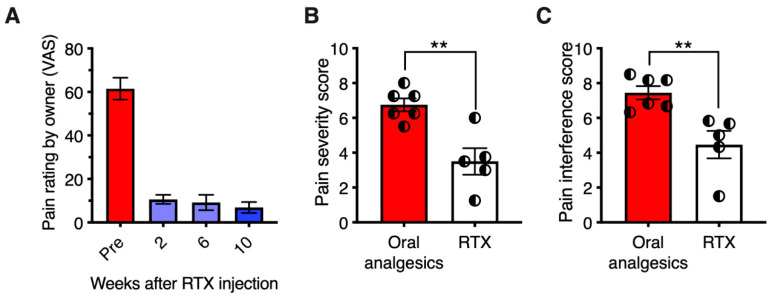
Treatment of osteosarcoma bone cancer pain and osteoarthritis pain with intrathecal RTX: efficacy and duration of action. The studies shown were conducted under approved protocols. (Panel (**A**)) Strong efficacy of a single administration of RTX was observed for treatment of bone cancer pain or osteoarthritis pain, or both, as rated by the owners in the companion canine model. A long duration of analgesia was also evident. An intracisternal dose of 1 μg/kg was administered under general anesthesia via cisternal puncture. Owners used a Visual Analog Scale to report pain severity. RTX produced a sustained reduction in pain in the eight dogs. The bars represent the average rating ± SEM. All post-treatment ratings were significantly different from pre-drug baseline (ANOVA with Scheffe’s post hoc test; *p* < 0.05). The animals initially presented with limb guarding as they walked, which improved over time, as did activity. The efficacy of RTX action was also evidenced by discontinuation of or reduction in supplementary analgesics (opioids and NSAIDs in all eight dogs) (data from [176], with permission). (Panels (**B**,**C**)) A separate group of 11 animals (6 standard-of-care with oral analgesics and 5 RTX 1.2 μg/kg intrathecally). Owners for each dog rated pain using the Canine Brief Pain Inventory. Pain Severity (**B**) and Pain Interference (**C**) were significantly decreased in the RTX group relative to dogs on standard-of-care oral analgesics alone. Statistical comparisons were made using two-tailed Mann–Whitney U test (**, *p* < 0.01). Figure courtesy of Matthew J. Sapio and Michael J. Iadarola; data are from [156], with permission).

## 7. Clinical Trials: Intrathecal or Epidural RTX for Permanent Analgesia in Cancer Patients

An open-label, single-site, Phase I clinical trial with increasing intrathecal RTX doses (NCT 00804154) was carried out under a Cooperative Research and Development Agreement between the National Institute of Neurological Disorders and Stroke (NiNDS), Bethesda, MD and Sorrento Therapeutics [178]. For this study, cancer patients with intractable cancer pain (for example, women with cervical cancer metastatic to the pelvic bone) were recruited. The protocol by and large followed that used in companion dogs with bone cancer pain. Four patients were placed under general anesthesia for one to two hours to minimize discomfort after RTX administration. RTX was injected manually over 2 min in a total volume of 1 mL. The first study participant experienced pain relief at the starting dose (3 µg), whereas the second patient needed a second dose of 13 µg. The third and fourth patient also received the 13 µg dose. Three additional patients were given a higher RTX dose of 26 µg. The higher RTX doses caused transient urinary retention and impaired noxious heat sensation as on-target side effects [179]. For example, a few patients suffered scalding injuries from grabbing hot coffee. This is consistent with the adverse effects of the pharmacological blockade of TRPV1 by small-molecule antagonists [22,23].

This small (a total of nine patients) study implies that intrathecal RTX may exert a clinically useful analgesic action in cancer patients, with manageable adverse effects. A few patients even became ambulatory after the RTX treatment. At present, this study is recruiting patients for testing the analgesic potential of the 44 µg RTX dose.

The intrathecal route of RTX administration requires general anesthesia. Furthermore, it affects both blood pressure and heart rate. Epidural RTX seems to be devoid of these complications [180].

The first-in-human epidural RTX study enrolled 17 participants and employed doses in the range of 0.4 µg to 25 µg given under mild sedation [181]. RTX was either injected directly into the epidural space or through a catheter placed by fluoroscopic guidance. The results of this trial are detailed elsewhere [181]. Briefly, three subjects achieved 30%, 50%, and 70% reductions in pain scores (calculated for both average and worst pain), respectively, lasting until the end of the study (12 weeks). Four patients withdrew from the study or were lost to follow-up, and another ten died due to the progression of the underlying cancer. Although this is a very limited study, it is promising that all the three patients who had completed the trial reported some degree of pain relief by epidural RTX. 

Sorrento Therapeutics plans to perform a randomized, blinded, placebo, and standard-of-care controlled Phase II trial with epidural RTX (15, 20 and 25 µg) in 120 patients with advanced cancer (NCT 05067257) [180,181].

## 8. Clinical Trials: Breakthrough Therapy Designation for Intra-Articular RTX to Treat Pain Associated with Knee Osteoarthritis

Osteoarthritis is the most common form of arthritis that predominantly affects the knees, hips, and hands. Osteoarthritis involves the whole joint, causing cartilage degradation, bone remodeling/osteophyte formation, and synovial inflammation [182,183]. Since osteoarthritis has a complex and only partially understood pathology, few targeted pharmacological treatment options are available.

The Swiss biotech company Mestex AG has developed an injectable RTX solution (MTX-071) to treat osteoarthritic knee pain. In a Phase IB clinical trial, 40 patients (35 to 85 years old) with moderate-to-severe knee pain caused by advanced osteoarthritis received an RTX dose of 12.5 µg into the knee joint [184]. Side effects included injection site pain (100% in the RTX group compared to 80% placebo), mild nausea (43%), vomiting (17%), and headache (10%). Overall, the side effects were manageable and well tolerated [184]. 

In the numerical pain rating scale, 83% of the patients reported a significant, clinically meaningful 2.63 decrease in the WOMAC A1 score (*p* = 0.0311). In the one-year follow-up, three patients maintained full response [184]. 

On 12 April 2021, Grünenthal (a German pharmaceutical company) acquired Mestex AG. Grünenthal is ready to start a Phase III clinical trial at multiple sites, involving 1800 patients [20,185]. For this study, the US Food and Drug Administration has granted “breakthrough therapy” status [5]. 

Importantly, similar relief of osteoarthritic pain was reported by Sorrento Therapeutics (San Diego, CA, USA) following intra-articular RTX injections (Figure 6).

## 9. Innovative RTX Uses: Beyond Bladder Control and Pain Relief

An interesting and controversial clinical use of topical RTX is in treating life-long premature ejaculation [186,187,188].

In rats, intra-gastric RTX blocks both basal and stimulated acid secretion [189]. Moreover, intra-gastric RTX (0.38 to 6.1 µM) protects animals against indomethacin or ethanol-induced ulcer formation [190,191]. In GERD (gastro-esophageal reflux disease) models, RTX limits the mucosal damage [192]. In the colon, intraluminal RTX reduces the severity of *Clostrium difficile*-associated pseudomembranous colitis [193]. In colitis models, RTX blocks anxiety- and depression-like behavior [194]. 

In periodontitis-susceptible Fischer rats, RTX inhibits experimental periodontitis [195]. A similar effect was noted in a murine model of periodontitis after intra-ganglionic RTX injection [196].

Chemoablation by RTX of cardiac afferents protects the rat heart from pressure overload-induced cardiac hypertrophy [197]. Intrathecal RTX reduces ventricular arrhythmia during heart failure [198]. In dogs, microinjection of RTX into the stellate ganglia protects against ischemia-induced ventricular arrhythmias [199].

In dogs and ferrets, RTX exerts a marked antiemetic action [200,201].

Lastly, RTX was promoted to improve the prognosis of COVID patients [202].

## 10. Conclusions and Future Research Directions

RTX is an ultrapotent capsaicin analog with a broad therapeutic window, allowing for the full desensitization of TRPV1-expressing sensory afferents by means of a single injection [69]. RTX desensitization does not affect mechanical sensation, motor coordination, or proprioception. It also spares higher brain functions [81]. 

Desensitization to RTX is by definition reversible [69]. Intravesical RTX was shown to restore continence in a subset of patients with idiopathic and neurogenic detrusor overactivity [203,204,205]. Although case reports indicated a therapeutic value for intravesical RTX in intersitital cystitis patients [138,139,140], larger controlled clinical studies failed to support these reports [141]. 

RTX is a highly lipophilic molecule. Keeping RTX bioavailable in aqueous solutions is challenging. The lack of commercially available intravesical RTX preparations hinders clinical use. Nonetheless, it was stated that the “future of bladder control is a pinch of pepper and gene therapy” [206]. The challenge is to identify the patients who will benefit from this “pinch of pepper.” An intravesical capsaicin test or TRPV1 expression in urothelium may guide this decision [106,144]. Those who do not respond to capsaicin or lack urothelial TRPV1 expression should be excluded from the clinical studies. 

RTX can also ablate sensory neurons, causing permanent analgesia [175,176,177]. In fact, RTX was referred to as a “molecular scalpel” to perform precision surgery on the pain pathway [177]. This targeted (intrathecal or epidural) RTX therapy holds great promise in cancer pain management [180]. The epidural approach seems to offer comparable pain relief with fewer side effects [180]. This needs to be confirmed in larger clinical trials. In cancer patients with intractable pain, RTX may alleviate or at least reduce the need for opioids. 

Intra-articular RTX has been given “breakthrough therapy designation” by the US Food and Drug Administration (FDA) to treat moderate-to-severe knee pain in patients with osteoarthritis [5]. Similar targeted approaches may be useful in the management of post-operative pain [153] or pain associated with severe burn injuries [154]. 

Animal experiments imply therapeutic utility for site-directed RTX administration in various disease states like periodontitis [195,196]. These indications are yet to be tested in human patients. Periodontitis affects ~20% of the global adult population, representing more than one billion cases worldwide [207]. In the elderly, the prevalence of gum disease exceeds 70% [208]. This is an important public health problem: it is the main cause of teeth loss. In the US, the average cost of periodontal treatment ranges from USD 1500 (scaling) to USD 8000 (if surgery is needed) [209]. Topical RTX may represent an inexpensive alternative to dental intervention to prevent or treat gum disease.

Most recently, a polymer-coated nanoparticle cream formulation of RTX, Resinizin (Ion Channel Pharma, Noblesville, IN) [210], has been reported to alleviate thermal hyperalgesia in a rat model of diabetic neuropathy [211]. The topical RTX treatment was also effective in mini pigs. Importantly, the Resinizin cream was devoid of the intense initial burning sensation that hinders the use of high-concentration capsaicin creams [211]. If these preclinical observations are confirmed in humans, topical RTX creams may represent a novel analgesic approach in patients with painful peripheral neuropathy.

## Figures and Tables

**Figure 1 ijms-24-15042-f001:**
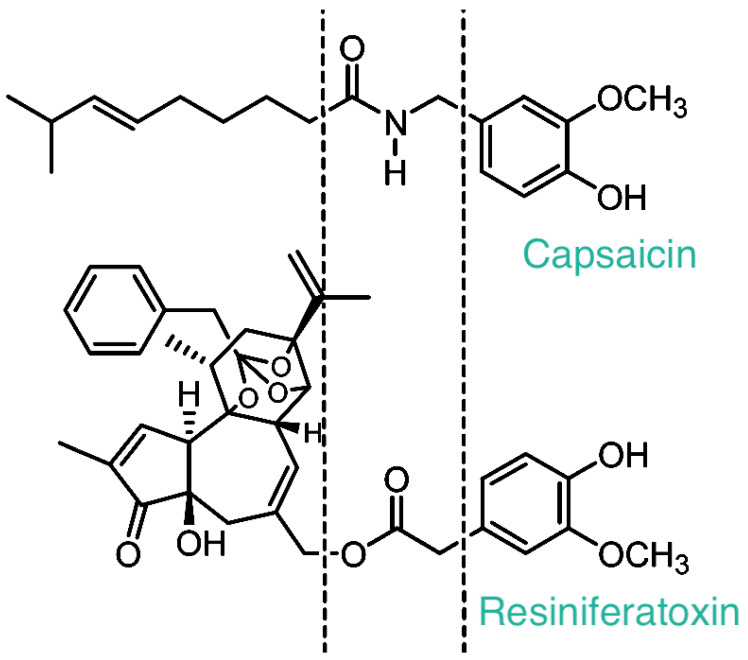
The chemical structure of capsaicin and resiniferatoxin. Please note the homovanillyl group shared by these two irritant compounds. The receptor that recognizes both capsaicin and resiniferatoxin (RTX) was named after this moiety as the “vanilloid” receptor (now known as TRPV1).

**Figure 2 ijms-24-15042-f002:**
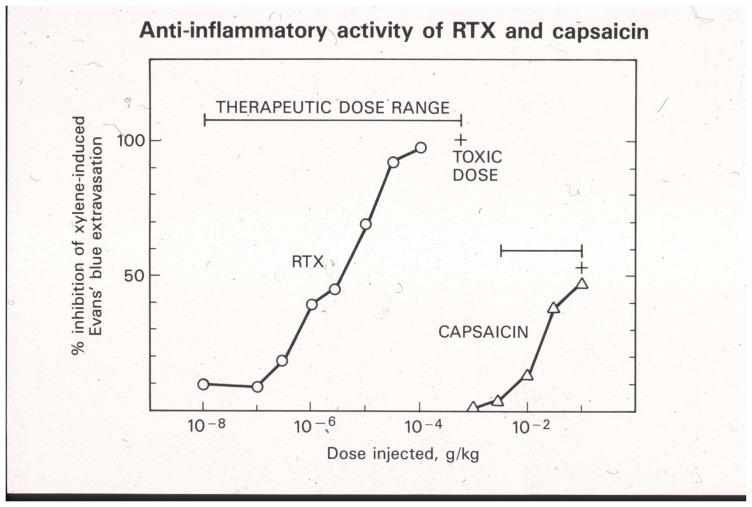
Desensitization of sensory neurons by systemic (s.c.) RTX and capsaicin administration blocks the xylene-induced neurogenic inflammatory response in the adult rat. Please note the differences between capsaicin and RTX actions: (1) RTX is at least a thousand-fold more potent than capsaicin, and (2) full desensitization can be achieved by a single RTX injection, whereas in the same protocol, capsaicin can only produce partial desensitization. Reproduced with permission from [11].

**Figure 3 ijms-24-15042-f003:**
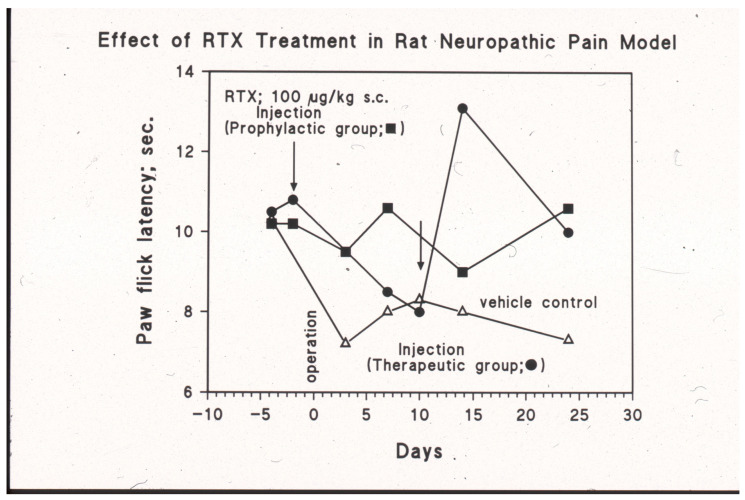
Systemic (s.c.) RTX injection (100 µg/kg) ameliorates thermal hyperalgesia in the Bennett model of neuropathic pain. The pain behavior was initiated by placing loose ligatures on the sciatic nerve of the rat (day 0, operation). Given three days before the operation (the “prophylactic” group), RTX prevents the development of the thermal hyperalgesia (closed squares). In the vehicle control group (open triangles), full thermal hyperalgesia occurs at day 10 after the surgery. In this group, “therapeutic” RTX rapidly reverses the pain behavior (closed circles). M. Tal and A. Szallasi, unpublished experiments.

**Figure 4 ijms-24-15042-f004:**
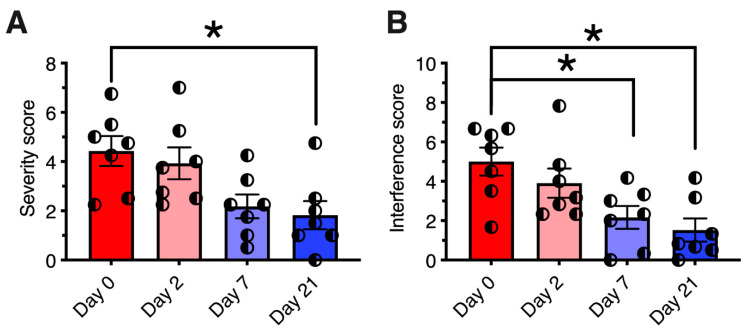
Analgesic effect of resiniferatoxin (RTX) in canine osteoarthritis. Seven dogs with osteoarthritis were treated under an approved protocol with a single intraarticular injection of 10 mcg of resiniferatoxin (RTX). (Panel (**A**)) Pain severity scores are plotted for each individual dog treated with RTX. (Panel (**B**)) Pain interference scores, same treatment. The Canine Brief Pain Inventory pain severity scores showed a significant reduction at day 21 post-treatment relative to pre-treatment values. Bars show the mean score, and half-filled circles represent values for each individual animal. *, *p* ≤ 0.05. E. Analgesic action was sustained for a median time of 150 days (range 58–730 days). Figure courtesy of Matthew J. Sapio and Michael J. Iadarola; data are from [156], with permission.

**Figure 6 ijms-24-15042-f006:**
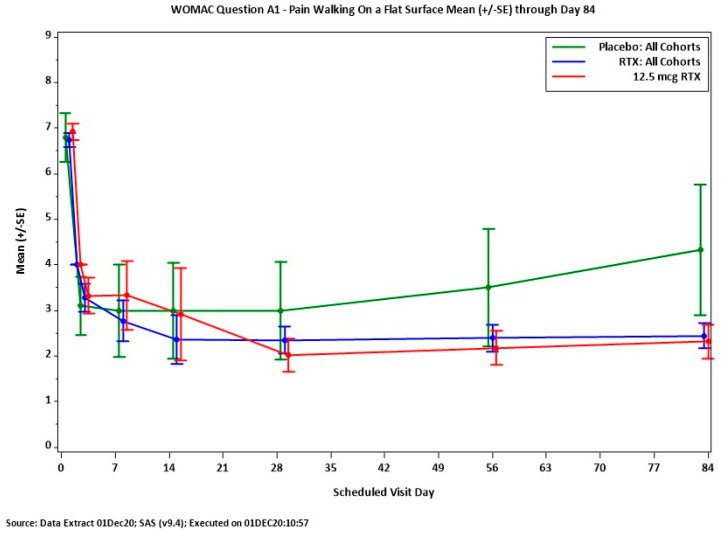
Analgesic action of intra-articular RTX (12.5 mcg) in patients with moderate-to-severe osteoarthritic pain. WOMAC Question 1, pain felt when walking on a flat surface. Between day 7 and 84 after treatment, RTX provides pain relief superior to that of placebo. Figure courtesy of Dr. Mike Royal (Sorrento Therapeutics).

**Table 1 ijms-24-15042-t001:** Potency of resiniferatoxin relative to that of capsaicin: representative examples.

Hypothermia in rats	7000	[11]
Provoking neurogenic inflammation in rats	1000	[11]
Blocking neurogenic inflammation in rats	20,000	[11]
Pungency in the eye-wiping assay	10	[11]
Increase in tail-flick latency	1000	[16]
Blocking acetic acid-induced writhing	6000	[16]
Bradycardia in the cat	60	[17]
Depressor reflex in rabbit ear	3	[18]
Ca^2+^ uptake in culture DRG neurons	100	[19]
Contraction in rat urinary bladder	1	[18]
Desensitization of rat urinary bladder	1000	[18]

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
