# Peer review of "Resiniferatoxin: Nature’s Precision Medicine to Silence TRPV1-Positive Afferents"

_ijms, 2023, doi:10.3390/ijms242015042_

Round 1
Reviewer 1 Report
This is a well written, well-researched up-to-date review. It is elegant by a broad and historic introduction, covers a wide array of potential uses and the cellular and animal data to this, and ends with the current progress in getting RTX into clinical use. I endorse publication of this knowledgeable work and have only some minor points:
1) There are many examples listed throughout the body, where ablation of TRPV1-sensitive afferents seem beneficial. The reader might be left puzzled, whether we should all get neonatal ablation of all of these fibers. Somewhere in the review, adding a few words what or in which conditions theses neurons are actually good for would seem helpful.
2) The ‘conclusion’ section can be shortened, as it is rather a summary of the points before than many conclusion drawn from that.
3) At several times, the lack of EM changes by RTX is mentioned, while EM changes particular to mitochondria are a hallmark of capsaicin damage. Beyond mentioning this, the reader would benefit from results or even hypothesis/speculation, why the EM findings are different for these two agonists.
4) Triphasic response: CGRP is not mentioned; do you concluded that CGRP released from peptidergic fibers plays no role?
5) Summary sentence on intravesical use. The difference in data quality between neurogenic and interstitial inflammation is nicely elaborated on in detail. However, the summary sentence does not make this distinction.
6) Line 336: This is rather selective, what happened in the other 14 participants.
7) Line 384: Quite similar, the reader will not understand why no grand mean with significant difference or not is reported, but only the outcome of 83% of the study population
Grammatical:
Line 90 ‘In the plasma membrane …’
Line 131: this should be 'systemically', not 'systematically'
Line 386: the commas surrounding Grünenthal makes ist sound as if Grünenthal was the only pharmaceutical company in this country
Reviewer 2 Report
See attached.

There were just a few places that needed clarification.
Reviewer 3 Report
In the present manuscript, the authors described the pharmacological importance of RTX and its significance in pain management through receptor TRPV1. They also compare the effect of RTX along with capsaicin. However, no review methodology and rationale of study is included. The information regarding RTX is included but why they include the capsaicin in study?? If they are targeting only the RTX, then they should include it only why capsaicin is only included not any other similar product. It is better to include the tables indicating the significant effect of RTX and detailed diagrams with respect to pharmacology of RTX.
Further, the review must include a discussion section, where authors should discuss the importance of selected topic and compiled information. Review is not only the compilation of information without adding the significant contribution by the authors. The authors should focus on the prospects of study with proper justification. For example, In conclusion section the authors writes that animals experiments imply therapeutic utility for site-directed RTX administration in various disease states, likeperiodontitis and gastric ulcer , cardiac arrhythmias [175-177] etc and these indications are yet to be tested in human patients. The authors should justify such suggestions with proper citations that why such studies could be useful in humans.
Moderate editing is required.
Round 2
Reviewer 3 Report
The authors provided a very brief response to the queries raised by the reviewer. It is further suggested to elaborate the response with point wise justification to the points like
- Inclusion and exclusion criteria of the 1062 paper to 200.
- Why the tables and diagrams are not included as suggested? These are must for the article.
- Discussion is not presented as suggested. The authors should focus on these important points and elaborate these points in the authors reply also.
Further it is also suggested to include the word Capsaicin in the title of manuscript as authors also focus on this plant active in the article.
Minor editing is required
Round 3
Reviewer 3 Report
The authors did not improve the manuscript as suggested.
Minor english editing is required